# A neural multilevel method for high-dimensional parametric PDEs

**Cosmas Heiß**[*]
Technical University of Berlin
cosmas.heiss@gmail.com

**Ingo Gühring**[*]
Technical University of Berlin
guehring@math.tu-berlin.de

**Martin Eigel**
Weierstraß Institute
martin.eigel@wias-berlin.de

## Abstract

In scientific machine learning, neural networks recently have become a popular tool for learning the solutions of differential equations. However, practical results often conflict the existing theoretical predictions in that observed convergence stagnates early. A substantial improvement can be achieved by the presented multilevel scheme which decomposes the considered problem into easier to train sub-problems, resulting in a sequence of neural networks. The efficacy of the approach is demonstrated for high-dimensional parametric elliptic PDEs that are common benchmark problems in uncertainty quantification. Moreover, a theoretical analysis of the expressivity of the developed neural networks is devised.

## 1 Introduction

The application of current machine learning methods to problems based on mathematical models has become one of the most promising rising research areas. In engineering and the natural sciences in particular, complicated mathematical models (usually formulated in terms of differential equations) form the basis for new insights and groundbreaking technologies. Since the computation of such models can be extremely time consuming, the use of machine learning tools was obvious and has become increasingly common in recent years[2]. Consequently, significant progress has been made on the one hand to understand the expressivity requirements for representing solutions of differential equations [17, 23, 5, 19, 14]. On the other hand, a plethora of architectures has been devised to implement differential equation solvers [4, 21, 24, 25]. Interestingly, while analytical results predict a superior capacity of deep neural networks e.g. for the important cases of partial differential equations (PDE) and stochastic differential equations (SDE), observed results unfortunately do not reflect the theory [1]. Instead, the convergence of the error typically stagnates after some training iterations as e.g. illustrated in [13, 16]. In particular, it often does not reach the accuracy of "classical" best-in-class methods (e.g. stochastic Galerkin or collocation, tensor regression, compressed sensing).

In the following, we are concerned with learning the solution map $v : \mathbb{R}^p \to H_0^1(D)$ of linear parametric PDEs of the form

$$\mathcal{D}(v, \mathbf{y}) := f - A(\mathbf{y})v(\mathbf{y}) = 0, \tag{1}$$

defined with a linear differential operator $A$, fixed right-hand side $f$, appropriate boundary conditions and a $p$-dimensional parameter vector $\mathbf{y} \in \mathbb{R}^p$ (determining the model data) on some domain

---

[*]Co-first author

[2]this is often coined scientific machine learning (SciML)

35th Conference on Neural Information Processing Systems (NeurIPS 2021), Sydney, Australia.

$D \subset \mathbb{R}^d$. Such (possibly very high-dimensional) PDEs are common problems in the field of Uncertainty Quantification (UQ). They can be used as meaningful benchmark problems to assess the quality of novel SciML methods. Note that in contrast to many classical machine learning problems, a dataset can be generated on the fly with arbitrary precision using the Finite Element (FE) method.

We recollect the notion of multilevel methods that are ubiquitous in numerical methods. These consist of a hierarchical construction of the approximation, relying on a set of successively refined discretizations and hence exploiting a decomposition of the problem complexity. Notably, for our method this means that in each approximation step, only the correction to the next finer level has to be represented in the respective sub-network.

The method which comes closest to our work was recently published in [18]. The three main differences are: a) We train the networks on each grid level to also correct the errors of the previous predictions (which turns out to be a significant design choice). b) We learn the solution $v(\mathbf{y})$ of the parametric PDE instead of some quantity of interest. For this to be feasible we make use of a compression of the FE basis. c) We examine the required theoretical network complexity and give an evaluation on standard UQ problems. Our numerical examples illustrate that the multilevel construction leads to a stable convergence of the parametric solutions which is comparable to best-in-class methods such as the ones in [9, 11, 12, 10, 6, 2].

## 2  A neural multilevel method

To construct the multilevel scheme for the considered PDE, we denote by $\mathcal{V}_1 \subset \ldots \subset \mathcal{V}_L \subset H_0^1(D)$ a nested sequence of FE spaces resulting from $L$ successive (uniform) refinements of a coarse grid on the spatial domain $D$ and let $v_\ell \colon \mathbb{R}^p \to \mathcal{V}_\ell$ be the function which maps a (stochastic) parameter $\mathbf{y} \in \mathbb{R}^p$ to the FE solution $v_\ell(\mathbf{y})$ of (1). Note that we can express the solution on the finest mesh $v_L$ as a sum of corrections

$$v_L(\mathbf{y}) = v_1(\mathbf{y}) + \sum_{\ell=2}^{L}(v_\ell(\mathbf{y}) - v_{\ell-1}(\mathbf{y})) =: \sum_{\ell=1}^{L} \tilde{v}_\ell(\mathbf{y}), \quad \text{where each} \quad \tilde{v}_\ell(\mathbf{y}) \in \mathcal{V}_\ell.$$

Since $\dim \mathcal{V}_\ell$ grows exponentially with $\ell$, we make use of a proper orthogonal decomposition (POD) [3, 22, 20] to yield spaces $\mathcal{U}_\ell = \mathrm{span}\{u_\ell^1, \ldots, u_\ell^{\dim \mathcal{U}_\ell}\}$ of lower dimension which allow an efficient approximation of the corrections $\tilde{v}_\ell(\mathbf{y})$.

In our neural multilevel method, we use neural networks $\mathcal{NN}_\ell \colon \mathbb{R}^p \to \mathbb{R}^{\dim \mathcal{U}_\ell}$ for $\ell = 1, \ldots, L$ to learn the basis coefficients $c_\ell^i(\mathbf{y})$ of the approximate subspace corrections

$$v_L(\mathbf{y}) = \sum_{\ell=1}^{L} \tilde{v}_\ell(\mathbf{y}) \approx \sum_{\ell=1}^{L} \sum_{i=1}^{\dim \mathcal{U}_\ell} c_\ell^i(\mathbf{y}) u_\ell^i \approx \sum_{\ell=1}^{L} \sum_{i=1}^{\dim \mathcal{U}_\ell} [\mathcal{NN}_\ell(\mathbf{y})]_i u_\ell^i.$$

To avoid an accumulation of the individual errors of $\mathcal{NN}_\ell$ in the above sum, we train $\mathcal{NN}_\ell$ to also correct the errors of the output of the preceding levels (see Step 2 below for a precise description). For this, in addition to the parameter vector $\mathbf{y}$, we also feed the output coefficients of all previous networks as input. We want to point out that correcting preceding errors provides an essential improvement over the method in [18] (see Section 3). For simplicity, in the following we just write $\mathcal{NN}_\ell(\mathbf{y})$ for the output vector of the $\ell$-th network (instead of $\mathcal{NN}_\ell(\mathbf{y}, \mathcal{NN}_{\ell-1}, \ldots, \mathcal{NN}_1)$).

To generate the training data, we draw $N \in \mathbb{N}$ samples $\mathbf{y}_1, \ldots, \mathbf{y}_N$ and compute solutions $v_\ell(\mathbf{y}_1), \ldots, v_\ell(\mathbf{y}_N)$ for each $\ell = 1, \ldots, L$ using the FE method. The iterative training procedure of our neural multilevel procedure reads now as follows:

**Algorithm 2.1.**

***Step 1:*** *(i)* *Compute the POD basis functions $u_1^1, \ldots, u_1^{\dim \mathcal{U}_1}$ from the snapshot matrix of FE solutions $[v_1(\mathbf{y}_1), \ldots, v_1(\mathbf{y}_N)]$ subject to a desired accuracy.*

*(ii)* *Train the first network $\mathcal{NN}_1$ to minimize*

$$\sum_{k=1}^{N} \left\| \left( \sum_{i=1}^{\dim \mathcal{U}_1} [\mathcal{NN}_1(\mathbf{y}_k)]_i u_1^i \right) - v_1(\mathbf{y}_k) \right\|_{H_0^1(D)}^2 .$$

***Step 2:*** *For $\ell = 2, \ldots, L$:*

    *(i) Successively update the current approximation*

$$w_{\ell-1}(\mathbf{y}_k) := \sum_{j=1}^{\ell-1} \sum_{i=1}^{\dim \mathcal{U}_j} [\mathcal{NN}_j(\mathbf{y}_k)]_i u_j^i.$$

    *(ii) Compute the POD basis functions $u_\ell^1, \ldots, u_\ell^{\dim \mathcal{U}_\ell}$ from the snapshot matrix of corrections $[v_\ell(\mathbf{y}_1) - w_{\ell-1}(\mathbf{y}_1), \ldots, v_\ell(\mathbf{y}_N) - w_{\ell-1}(\mathbf{y}_N)]$.*

    *(iii) Train the network $\mathcal{NN}_\ell$ to approximate the correction by minimizing*

$$\sum_{k=1}^{N} \left\| \sum_{i=1}^{\dim \mathcal{U}_\ell} [\mathcal{NN}_\ell(\mathbf{y}_k)]_i u_\ell^i - (v_\ell(\mathbf{y}_k) - w_{\ell-1}(\mathbf{y}_k)) \right\|_{H_0^1(D)}^2.$$

## 3 Numerical examples

To assess the performance of Algorithm 2.1, we consider a common high-dimensional PDE and compare the solution approximation of our neural multilevel scheme with two related NN models. More specifically, we use the stationary parametric diffusion problem $-\mathrm{div}(T(a(\mathbf{y}))\nabla u(\mathbf{y})) = 1$ with homogeneous Dirichlet boundary conditions on the unit square. The conductivity is given by $a(y) = \mu + \sum_{m=1}^{p} a_m \mathbf{y}_m$ where $a_m \sim m^{-2}$ are planar Fourier modes, cf. [9]. We consider the "uniform case" with $T = \mathrm{id}$, $\mu = 1$, $p = 50$ and $\mathbf{y}$ drawn from $U([-1,1]^p)$, as well as the severely more involved "log-normal case" with $T = \exp$, $\mu = 1$, $p = 50$ and $\mathbf{y}$ drawn from $N(0, I)$. Algorithm 2.1 is employed with $L = 7$ feed-forward ReLU neural networks $\mathcal{NN}_1, \ldots, \mathcal{NN}_L$ consisting of two hidden layers and $512$ nodes per layer.

We analyse the approximation quality of our neural multilevel scheme by comparing it with the closely related architecture of [18] (denoted "Lye et al.")[3] on an independent test set consisting of $1024$ randomly drawn samples. Moreover, we compare the results to a single larger feed-forward ReLU model (denoted "Single Net") with roughly the same parameter count as all the multilevel networks of one architecture combined. "Single Net" is trained to directly predict an approximation on the finest mesh. For all three approaches, a POD is utilized with a prescribed tolerance of $10^{-7}$. The network training is based on $10^5$ samples that are randomly drawn according to the distribution of the respective test case. On the finest level 7, the mesh consists of about $2 \times 10^5$ triangles with first order Lagrange elements.

In Figure 1, the $H_0^1$-error with respect to a high fidelity reference solution of our method and "Lye et. al" is displayed for an increasing number of levels (solid line). It is noteworthy that our "ML Net" essentially keeps its convergence rate throughout while "Lye at al." flattens out, starting at level 4 or 5, respectively. We also reach a relative error that is much better than what is shown in other NN methods such as e.g. [13]. The reason for this may be found in the subtle difference of the two methods. "Lye et. al" predicts the level corrections independently of the neural network approximations of the previous level (and their errors) leading to a successive error accumulation from level to level. In contrast, on each level, "ML Net" is aware of the previous errors and trained to correct them. The $H_0^1$-error with respect to a high fidelity reference solution projected onto the corresponding grid (dotted line) supports this hypothesis. While the error for "ML Net" stays relatively constant with increasing levels we can observe an error accumulation for "Lye et al". The approximation error of the "Single Net" with respect to the high fidelity reference solution is indicated by a red triangle. Its performance is comparable to "Lye et al".

The results suggest that it is advantageous to learn the corrections iteratively with respect to the preceding approximation to mitigate error accumulation instead of learning the grid corrections independently. Throughout the experiments, our method is the only one whose approximation error is not significantly larger than the FE grid approximation error.

---

[3]We point out, that the original method from [18] aims at predicting quantities of interest. As it is the closest method to ours found in the literature, we adapted their approach to the setting considered in this paper.

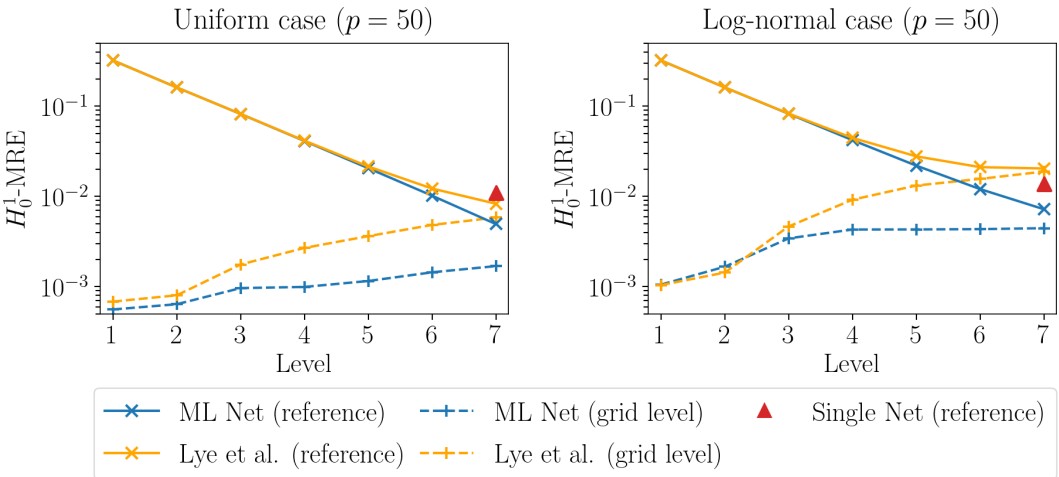

Figure 1: Relative energy errors evaluated on an independent test set for the uniform case with $p = 50$ (left) and the log-normal case with $p = 50$ (right).

## 4 Theoretical analysis

In this section, we analyse the complexity (in terms of the number of weights) of the neural networks in our multilevel method[4]. The proof is based on the well-established ability of neural networks to approximate polynomials efficiently [15, 8, 26] and will be published in our upcoming paper. Our result is in line with the literature [7, Section 3.9] and shows that for fixed stochastic dimension $p$, to reach the accuracy of the FE solution on the finest grid ($\sim \eta^{-L}$) it suffices to have a polynomially growing number of weights assuming a suitable dimensionality reduction. As expected, this rate deteriorates with increasing stochastic parameter dimension. A related analysis without using a multilevel scheme is presented in [23, 17].

**Theorem 4.1.** *For each $L \in \mathbb{N}$ there exist neural networks $\mathcal{NN}_1, \ldots, \mathcal{NN}_L$ where the total number of weights is bounded by*

$$\sum_{\ell=1}^{L} \#\mathcal{NN}_\ell \lesssim L^{2p+2} + L^p \sum_{\ell=1}^{L} \dim \mathcal{U}_\ell,$$

*such that the multilevel Algorithm 2.1 together with a POD of accuracy $\varepsilon_{POD}$ achieves the precision on the order of the FE solution on the finest grid (up to a multiplication of $L$)*

$$\left\| \sum_{\ell=1}^{L} \sum_{i=1}^{\dim \mathcal{U}_\ell} [\mathcal{NN}_\ell(\mathbf{y})]_i u_\ell^i - v(\mathbf{y}) \right\|_{H_0^1(D)} \leq \left( 2L\eta^{-L} + \varepsilon_{POD} \right) \|f\|_*.$$

Analysing Single Net in a similar way, we can show that its required complexity grows asymptotically faster than the bound in Theorem 4.1. This crucial observation shows the efficiency of the multilevel ansatz in terms of its approximative power.

## 5 Conclusion

We describe and numerically demonstrate a neural multilevel scheme which is comparable to best-in-class performance for the solution approximation of high-dimensional parametric PDEs as commonly encountered in UQ. As such, the proposed scheme outperforms other neural network approaches from the recent literature. Regarding analytical aspects, we provide complexity bounds which behave favourably with respect to the expected theoretical convergence for this type of problem. Moreover, asymptotic estimates also show the advantage of the proposed multilevel structure. For detailed theoretical results, we refer to our upcoming paper.

---

[4]For the analysis to be tractable, we ignore the dependence of the corrections on the output of the previous levels (see Algorithm 2.1 Step 2).

Nevertheless, despite the favourable observed performance, there are still several open questions. In particular, a theoretical non-asymptotic multilevel advantage as in other UQ methods has not been uncovered yet. Moreover, the required POD reduces the scalability of the approach and prevents an application when regularity of the parameter to solution map is low. In future work, one might consider different architectures to lessen the dependence on dimensionality reduction methods using e.g. CNNs.

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
