# OpenReview forum: "A neural multilevel method for high-dimensional parametric PDEs"
_NeurIPS.cc/2021/Workshop/DLDE — DLDE Workshop -- NeurIPS 2021 Poster_

### Official Review · Reviewer_K5p3 · 2021-10-02
**Solid works**

**Confidence:** 3

**Review:**

I think this article is written well. It contains both the empirical
study and the theoretical analysis, which are quite solid. The proposed method used neural
network to learn the basic coefficients in the multi-level scheme,
which works well for a stationary parametric diffusion problems.
To further demonstrate the feasibility of the proposed algorithm, I suggest that more extensive experiments including some other forms of PDE can be done in the future.
By the way, I am not an export on multi-level method in numerical analysis.
So I am confused when encountered with the jargon "FE basis".
I hope the author could give its full name to easy the confusion of
researchers.

Some grammar problems are listed as follows:
Line 39: We also train the networks to correct..
Line 135: "such as" cannot be used together with "e.g."


**Score:**

4: Very good paper

---

### Official Review · Reviewer_8Qts · 2021-10-03
**Interesting idea, but the main advantage of multi-level methods is not fully leveraged.**

**Confidence:** 5

**Review:**

The submission proposes a multi-level method to increase the accuracy of machine learning approach for solving high-dimensional parametric PDEs. Based on my understanding of this paper and recent prior work (ref. [18]), the contribution is relatively minor here.

More detailed concerns:
1) The main advantage of multi-level approach is to reduce the computational cost of standard sampling methods by generating most training samples with low accuracy at a corresponding low cost, whereas relatively few samples are taken at high accuracy and a high cost. However, due to the fact that a large number of samples is required to ensure the accuracy of POD reduced model at all levels, sample size of the proposed method does not vary along with the refinement of mesh resolution (see line 63-64). As such, the main advantage of multi-level method is not fully leveraged.

2) Though the performance of proposed algorithm is better than that of high-fidelity reference method, the computational cost of the former is also larger than that of the latter. For fairness, the comparison should be made under the same or similar computational cost.

3) The generalization ability of trained networks is not studied or reported, making the experimental section look very preliminary.

4) There is a gap between the proposed method and its corresponding theoretical analysis. To be specific, the network on each grid level is trained to correct the errors of its previous predictions, but the theoretical analysis ignores the dependence of the corrections on the output of the previous levels.

**Score:**

2: Borderline paper

---

### Official Review · Reviewer_Zyky · 2021-10-11
**Novel Multi-Level architecture for solving elliptic PDEs**

**Confidence:** 3

**Review:**

The authors propose solving differential equations using neural networks that make predictions at multiple successive levels of refinement of a Finite Element (FE) grid. The authors claim that at each step of refinement optimizing with respect to the error in preceding steps as well as the current one, provides gains with regards to convergence to solutions of elliptic PDEs. The authors re-parameterize the outputs of the network to this effect.
The authors empirically demonstrate that this is beneficial for convergence of the algorithm. The authors state a theorem that characterizes the lower bound (but do not provide a clear proof).

Pros:

The work demonstrates a novel contribution.

Experiments clearly highlight the benefits of the proposed method compared to other choices of architectures.

It would be interesting to see an empirical investigation of how the number of layers impacts the output and how this contrasts with the lower bound that is stated in the theorem.

Seeing more work on this method in the context of Uncertainty Quantification is promising.

Cons:

There's a lot of acronyms in the text that are not necessarily obvious to the reader. It would be better to explain these as they appear.

More details on the optimization procedure and how the error of each layer changes during training vs. the reference baseline would help further highlight the benefits of the proposed approach.  It would also be interesting to see how the size of the mesh impacts performance.




**Score:**

3: Good paper

---

### Decision · Program_Chairs · 2021-10-17

**Decision:**

Accept (Poster)

**Comment:**

This paper represents a novel contribution to the theory of multi-level approaches for solving to differential equations with deep neural networks. The work is a good contribution to the workshop and has been selected as a poster. The authors may consider this opportunity to address some of the limitations of the work that has been highlighted by reviewers.